Seed germination and vegetative and in vitro propagation of Hieracium lucidum subsp. lucidum (Asteraceae), a critically endangered endemic taxon of the Sicilian flora

Gianguzzi Valeria 1
Di Gristina Emilio 1 emilio.digristina@unipa.it
Barone Giulio 1 2
Sottile Francesco 3
Domina Gianniantonio 1 2
1 Department of Agricultural, Food and Forest Sciences (SAAF), University of Palermo , Palermo , Italy
2 NBFC, National Biodiversity Future Center, University of Palermo , Palermo , Italy
3 Department of Architecture, University of Palermo , Palermo , Italy
Serim Ahmet Tansel
Electronic publication date: 2024 Feb 9
Publication date: 2024
Volume: 12
Electronic Location ID: e16839
Received 2023 Sep 7; Accepted 2024 Jan 5
Copyright: © 2024 Gianguzzi et al.
Copyright year: 2024
Copyright holder: Gianguzzi et al.
License: This is an open access article distributed under the terms of the Creative Commons Attribution License, which permits unrestricted use, distribution, reproduction and adaptation in any medium and for any purpose provided that it is properly attributed. For attribution, the original author(s), title, publication source (PeerJ) and either DOI or URL of the article must be cited.
License URL: https://creativecommons.org/licenses/by/4.0/

Keywords: Conservation, Endemism, Leaf explants, Mediterranean flora, Sexual reproduction, Vegetative multiplication

Funding: NBFC to the University of Palermo Italian Ministry of University and Research, PNRR, Missione 4 Componente 2, “Dalla ricerca all’impresa”, Investimento 1.4 CN00000033 Progetto di Ricerca di Rilevante Interesse Nazionale PLAN.T.S. 2.0–towards a renaissance of PLANt Taxonomy and Systematics University of Pisa 2017JW4HZK “Fondo per il Finanziamento della Ricerca di Ateneo” (FFR 2022 E. Di Gristina, University of Palermo) This work was supported by NBFC to the University of Palermo, funded by the Italian Ministry of University and Research, PNRR, Missione 4 Componente 2, “Dalla ricerca all’impresa”, Investimento 1.4, Project CN00000033, by the “Progetto di Ricerca di Rilevante Interesse Nazionale” (PRIN) “PLAN.T.S. 2.0–towards a renaissance of PLANt Taxonomy and Systematics” led by the University of Pisa, under the grant number 2017JW4HZK, and by “Fondo per il Finanziamento della Ricerca di Ateneo” (FFR 2022 E. Di Gristina, University of Palermo). The funders had no role in study design, data collection and analysis, decision to publish, or preparation of the manuscript.

==============================
Hieracium lucidum subsp. lucidum is a critically endangered endemic taxa of the Sicilian flora. It is a relict of the Tertiary period surviving on the cliffs of Monte Gallo (NW-Sicily). This research focused on finding the best protocols for seed germination and vegetative and in vitro propagation to contribute to ex situ conservation. Seed germination tests were carried out using constant temperatures of 15 °C, 20 °C and 25 °C in continuous darkness and an alternating temperature of 30/15 °C (16 h/8 h, light/dark). The seeds had no dormancy, and a high germination capacity (70–95%) was obtained at all tested thermoperiods. The possibility of vegetative propagation of the taxon was evaluated through the rooting capacity of stem cuttings treated or not treated with indole-3-butyric acid (IBA). All cuttings were treated with IBA rooted within 2 months, while only 50% of the untreated cuttings were rooted within a longer time. An efficient protocol for rapid in vitro propagation from leaf portions was developed. The response of explants was tested on hormone-free Murashige and Skoog (MS) basal medium and MS enriched with different types of cytokinins: 6-Benzylaminopurine (BAP) and meta-Topolin (mT) in combination with naphthaleneacetic acid (NAA) and 2,4-Dichlorophenoxyacetic acid (2,4-D) at the same concentration. The combination of mT (2 mg L−1) and 2,4-D (1 mg L−1) in the medium was the most effective and showed the highest percentage of callus induction and the mean number of regenerated shoots. The maximum rate of root regeneration and the maximum number and length of roots were obtained on hormone-free MS and MS enriched with IBA at concentrations of 1 mg L−1. From the results obtained, it can be concluded that H. lucidum subsp. lucidum can be successfully propagated using one of the tested techniques, subject to the availability of the material for reproduction.

Introduction

As is well known, Hieracium L. s. str. (Asteraceae) is a genus of vascular plants in which diplosporous agamospermy, hybridisation and polyploidy have a fundamental role (Krahulcová, Krahulec & Chapman, 2000; Chrtek, Mráz & Sennikov, 2006). Sexuality is very rare and confined to a few diploid species, mostly distributed in southern Europe (Chrtek, Mráz & Severa, 2004; Mráz & Zdvořák, 2019). This complicated reproductive biology has given rise to a very large number of morphotypes that are hardly distinguishable from each other, and they have been described as either subspecies or at the rank of species (Mráz, Chrtek & Kirschner, 2001). Many of the taxa described are punctual endemics and require specific actions of protection.

Due to their climatic conditions, southern Italy and, in particular, Sicily are suitable for only a few Hieracium relict taxa. However, over the past 20 years, in-depth taxonomic studies of this genus have resulted in several endemic taxa with a very local distribution. Additionally, little-known taxa have been rediscovered or reported as new to Italian flora (Table 1). Several studies on cariology (Di Gristina, Domina & Geraci, 2014, 2021), seed germination (Di Gristina, Varshanidze & Domina, 2020), conservation status (Di Gristina, Raimondo & Domina, 2022) and nomenclature (Di Gristina et al., 2012) have also been conducted on these taxa.

Table 1 List of the new Hieracium taxa described, rediscovered (*) or reported as new (**) to the Italian flora in the last 20 years for Southern Italy and Sicily and their distribution.

	Taxon	Distribution	Reference	
	H. aspromontanum Brullo, Scelsi & Spamp.	Endemic to Calabria	Brullo, Scelsi & Spampinato (2001)	
	H. barrelieri Gottschl., Raimondo, Greuter & Di Grist.	Endemic to Campania	Gottschlich et al. (2015)	
	H. busambarense Caldarella, Gianguzzi & Gottschl.	Endemic to Sicily	Caldarella, Gianguzzi & Gottschlich (2013)	
	H. hypochoeroides subsp. cilentanum Di Grist., Gottschl. & Raimondo	Endemic to Campania	Di Gristina, Gottschlich & Raimondo (2016a)	
	H. hypochoeroides subsp. montis-scuderii Di Grist., Gottschl., Galesi, Raimondo & Cristaudo	Endemic to Sicily	Di Gristina et al. (2013)	
	H. hypochoeroides subsp. lucanicum (Arv.-Touv.) Di Grist., Gottschl. & Raimondo	Endemic to Campania	Di Gristina, Gottschlich & Raimondo (2015b)	
	H. hypochoeroides subsp. peracutisquamum Di Grist., Gottschl. & Raimondo	Endemic to Basilicata	Di Gristina, Gottschlich & Raimondo (2015a)	
	H. jurassicum subsp. serrapretense Di Grist., Gottschl. & Scafidi	Endemic to Basilicata	Di Gristina, Gottschlich & Scafidi (2018)	
	H. pallidum subsp. aetnense Gottschl., Raimondo & Di Grist.	Endemic to Sicily	Gottschlich, Raimondo & Di Gristina (2013)	
*	H. pollinense Zahn	Endemic to Basilicata	Gottschlich, Scafidi & Di Gristina (2017)	
	H. racemosum subsp. lucanum Di Grist., Domina, Gottschl. & Scafidi	Endemic to Basilicata	Di Gristina et al. (2019)	
	H. racemosum subsp. pignattianum (Raimondo & Di Grist.) Greuter	Endemic to Sicily	Raimondo & Di Gristina (2004)	
	H. schmidtii subsp. madoniense (Raimondo & Di Grist.) Greuter	Endemic to Sicily	Raimondo & Di Gristina (2007)	
*	H. schmidtii subsp. nebrodense (Lojac.) Di Grist., Gottschl. & Raimondo	Endemic to Sicily	Di Gristina, Gottschlich & Raimondo (2016b)	
	H. terraccianoi Di Grist., Gottschl. & Raimondo	Endemic to Sicily	Di Gristina, Gottschlich & Raimondo (2014)	
**	H. umbrosum subsp. abietinum (Boiss. & Heldr.) Greuter	Basilicata; Balkan Peninsula	Gottschlich, Domina & Di Gristina (2017)	

Sicily, the largest island of the Mediterranean basin, is situated at the southern fringe of the distribution range of Hieracium. On this island, the genus is currently represented by 12 taxa, 11 of which are endemic.

Hieracium lucidum Guss. subsp. lucidum is a suffruticose chamaephyte endemic to Mt. Gallo (Fig. 1), a limestone promontory located near the city of Palermo (NW-Sicily). It is a chasmophyte strictly localised on the shady northwest cliffs, between 220 and 310 m a.s.l. These cliffs often benefit from the formation of fog due to the fresh and humid winds coming from the sea. These conditions allow the survival of other punctual endemic species, such as Anthemis ismelia Lojac., Genista gasparrinii (Guss.) C. Presl, Limonium panormitanum (Tod.) Pignatti and Pseudoscabiosa limonifolia (Vahl) Devesa. Although Mt Gallo is a regional natural reserve, it is affected almost yearly by wildfires that reach the vertical walls. The morpho-anatomical study of H. lucidum subsp. lucidum has made it possible to detect a structural organisation of leaves and stems that demonstrate a poor tolerance to the xerophilic environment of the mountain; it is markedly mesophilic, and its survival is allowed by the constant use of the non-rainfall water provided by the fog (Colombo, Melati & Princiotta, 1981). Therefore, H. lucidum subsp. lucidum is a paleoendemism, representative of the subtropical flora that occupied the Mediterranean area in the Tertiary period, which today finds refuge only on Mt. Gallo (Brullo & Brullo, 2020; Gianguzzi, Caldarella & Pasta, 2022). H lucidum subsp. lucidum is also one of the few sexual and diploid examples (Merxmüller, 1975; Brullo & Pavone, 1978; Brullo, Campo & Romano, 2004) in the genus Hieracium, and according to Pignatti (1979, 1982, 1994, 2018), it could be an ancestral taxon from which several European taxa may be derived. Although the plants grow on rocks and vertical cliffs, the recurrent presence of fire progressively reduces the size of its single population and according to Di Gristina, Raimondo & Domina (2022), H. lucidum subsp. lucidum can be classified as Critically Endangered (CR B1ab(iii,v)). Currently, the protection of the species is not provided for in any law or convention. Due to the current state of conservation, the phytogeographic relevance and its peculiar taxonomic role within Hieracium, the taxon requires immediate and special protection measures. Hence, there is a need to operate not only with fire control but also with planned population reinforcement interventions. Consequently, we initiated an extended work to develop efficient protocols for seed germination and vegetative and in vitro propagation. The protocols for seed germination and vegetative propagation for reproduction of Endangered Species of the Mediterranean area are quite developed (Magrini & Salmeri, 2022). To date, in vitro micropropagation studies aimed at species conservation are less common, and no studies are known about Hieracium taxa. To conserve genetic heritage, in vitro propagation can be used in ex situ conservation strategies. Indeed, establishing a micropropagation method for culturing that facilitates the rapid production of a substantial number of plantlets could make a significant contribution to the conservation of species. Germplasm conservation requires cooperation among a spectrum of different technologies (Falk, 1990; Fay, 1992). Micropropagation and other in vitro techniques have been widely used in conservation for the propagation of endangered species and species that present difficulties with traditional propagation techniques. Many threatened species can be quickly propagated with this technology using a minimum of plant material and reducing the impact on populations (Bengi & Yeald, 2005; Trejgell, Michalska & Tretyn, 2010; Gianguzzi et al., 2023).

Figure 1 Hieracium lucidum subsp. lucidum in its locus classicus.

The results concerning the three reproduction protocols reported in this article aimed to contribute to the conservation of this remarkable element of Sicilian flora.

Materials and Methods

Seed germination tests

The cypselae, hereafter called seeds, of Hieracium lucidum subsp. lucidum were collected during the autumn of 2020 and 2021 from plants growing in their locus classicus. For the collection, fine mesh nets were used to avoid seed dispersion and, at the same time, to allow the passage of light and air for the completion of maturation. After harvesting, the seeds were cleaned, and the pappi were removed manually and stored at room temperature.

Germination tests were carried out 15 days after seed harvesting according to Di Gristina, Varshanidze & Domina (2020), using a Petri dish with 20 seeds to test each variable (a total of 160 seeds per year). Due to the low number of mature seeds produced by the plants (about 10–15 per capitulum) and to prevent damage to the natural population, estimated at about 50 mature individuals, it was not possible to test a higher number of seeds. Germination tests were conducted in pure water and in a 10−3 M gibberellic acid (GA3) water solution in Petri dishes with two sheets of filter paper at the bottom. The germination test in pure water and in GA3 solution was compared to verify the occurrence of dormancy. In fact, gibberellins are involved in the natural process of breaking dormancy (Stowe & Yamaki, 1957).

The germination tests were conducted in a growth chamber, using constant temperatures of 15 ± 1 °C, 20 ± 1 °C and 25 ± 1 °C in continuous darkness and alternating temperatures of 30/15 ± 1 °C (16 h/8 h, light/dark with a light intensity of 150 µmoles/m2/s). These temperatures were chosen because they are in line with those recorded in the field during the germination period (October–December); alternate temperatures chosen represent the extremes of daily temperature variation that can be observed during this period. Given the limited number of seeds available, choices had to be made regarding the conditions of the germination tests. The tests at constant temperatures in continuous darkness simulate the conditions found in pockets of soil in rock cracks where the seeds of Hieracium lucidum subsp. lucidum germinate. The alternation of light and temperature is to simulate the condition of seeds that germinate on open ground without the protection offered by rocks. The seeds were counted every day and those with a radicle length of 1–2 mm were considered to have germinated. The same germination protocol was repeated for seeds harvested in the autumn of 2021.

The results of the germination tests were summarised in a table using the same common germination indices used in Di Gristina, Varshanidze & Domina (2020).

A binary logistic regression (being the dependent variable dichotomous), in the class of GLM, was used to test the effects of different temperatures and germination solutions (see File S1).

The full factorial experimental design (4 × 2) was fully replicated in 2 years (2020 and 2021) to consider any possible changes occurred in the field between the two harvesting years.

The analysis was performed using MINITAB© software.

Vegetative propagation test

During the winter of 2021, 13–18 cm long apical stem cuttings (Fig. 2) were collected from pot-grown H. lucidum subsp. lucidum plants in the dormant phase at the Department of Agriculture, Food and Forest Sciences of the University of Palermo. In total, 24 cuttings were collected. The cuttings were disinfected with 3% sodium hypochlorite for 5 s and then washed with sterile distilled water. The basal portion of the 12 cuttings was treated with a talc formulation of indole-3-butyric acid (IBA), and 12 control cuttings were not treated. Of the 12 cuttings treated with IBA, six were placed in a rooting medium consisting of 50% sand-perlite and 50% peat, while the remaining six were placed in a rooting medium consisting of 50% fine sand and 50% sawdust. The same was done for the control cuttings. The vegetative propagation test was carried out in a greenhouse without heating (10–20 ± 1 °C). The low number of samples studied did not allow statistical analysis to be carried out but gave an indication of the treatment to be suggested for vegetative reproduction protocols.

Figure 2 Apical stem cutting of Hieracium lucidum subsp. lucidum.

In vitro propagation test

The in vitro propagation test followed the following steps: collection of the plant material, sterilisation, passage in the culture medium, callus induction, shoot regeneration, rooting of the regenerated shoots and acclimatisation.

Collection of the plant material and surface sterilisation

The experiment was conducted at the micropropagation laboratory of the Department of Agricultural, Food and Forestry Sciences, at the University of Palermo. A total of 20 leaves from 10 individuals of Hieracium lucidum subsp. lucidum were collected in the field in mid-September. Initially, the leaves were kept under running water for 1 h to remove any traces of dust and stirred in water plus detergent for 30 min. Subsequently, to reduce the contamination caused by fungi and endogenous or exogenous bacteria, the leaves were sterilised under a vertical laminar flow hood in ethanol at 70% for 5 min and then immersed in a commercial solution of 15% bleach for 15 min. Finally, the leaves were rinsed in distilled water, cut into pieces of about 1 cm2, and placed in Petri dishes with plant growth regulators (PGRs).

Culture media and growth condition

The sterilised leaves were placed inside Petri dishes with basal medium with vitamins (Murashige & Skoog, 1962) enriched with 30 g L−1 sucrose and 7 g L−1 plant agar (Duchefa, Haarlem, Netherlands) as an agent and integrated with different concentrations of plant growth regulators, as indicated in Table 2. A pH of 5.8 was reached by adding 1M NaOH or 1M HCl solution. The Petri dishes were maintained at 25 ± 1 °C in a climatic growth chamber in the dark and then subjected to a photoperiod of 16 h light under a white fluorescent lamp with a photosynthetic photon flux density (PPFD) of 35 μmol m−1 s−1.

Table 2 Plant growth regulators used in the media to induce the regeneration of callus and shoots from Hieracium lucidum subsp. lucidum.

Medium	PGR	
BAP mgL−1	mT mg L−1	NAA mg L−1	2,4-D mg L−1	
MS0	0	0	0	0	
B1	1	0	1	0	
B2	2	0	1	0	
B3	1	0	0	1	
B4	2	0	0	1	
mT1	0	1	1	0	
mT2	0	2	1	0	
mT3	0	1	0	1	
mT4	0	2	0	1	
Note:

MS0, control, Murashige and Skoog basal medium, hormone free; B1, B2, B3, B4, medium with 6-Benzylaminopurine (BAP), at different concentration of 1 or 2 mg L−1 in combination with 1-Naphthylacetic acid (NAA) and 2,4-Dichlorophenoxyacetic acid (2.4-D) at the concentration of 1 mg L−1; mT1, mT2, mT3, mT4, medium with meta-Topolin (mT) at concentration of 1 or 2 mg L−1 in combination with NAA at the concentration of 1 mg L−1.

Callus induction

For callus induction, leaves were added to MS basal medium with vitamins (Murashige & Skoog, 1962) enriched with 30 g L−1 sucrose and 7 g L−1 plant agar, and containing various concentrations and combinations of 6-benzylaminopurine (BAP), meta-Topolin (mT), 2,4-Dichlorophenoxyacetic acid (2,4-D) and Naphthylacetic acid (NAA) (Fig. 3A). An MS medium without any hormones (MS0) was used as a control. The pH range of the plant tissue culture medium is essential for the development and growth of explants (Rashid et al., 2018). In this experiment, the pH of the medium was adjusted to 5.7. A total of 56 explants were used (seven replicates with eight explants). The culture of the callus was carried out in Petri dishes of 90 mm diameter with 30 mL of medium and kept in culture for 4 weeks, after which the effect of the different PGRs on callus production was evaluated (Figs. 3B and 3C).

Figure 3 In vitro callus regeneration produced by leaves of Hieracium lucidum subsp. lucidum.

(A) The callus that developed in the medium; (B and C) Different stages of indirect organogenesis in the MS medium integrated with 3 mg L−1 BAP and 1 mg L−1 NAA; (D–F) shoot development in the culture medium.

Shoot regeneration

To evaluate the percentage of regeneration of the shoots, the obtained callus was transferred to microbox vessels (125 × 65 × 80 mm) containing 80–100 mL of culture media (Figs. 3D and 3E). The callus was subcultured twice in fresh medium after 15 days (Fig. 3F). The regeneration of the shoots from the callus was evaluated after 7 weeks. All cultures were incubated in a growth climatic chamber at 25 ± 1 °C in the dark and then placed under a fluorescent white lamp with a PPFD of 35 μmol m−1 s−1 and a photoperiod of 16 h of light and 8 h of darkness.

Rooting of the regenerated shoots and acclimatisation

To promote root development and further plant growth, the regenerated shoots were cut from the callus and transferred to hormone-free MS basal medium with vitamins (Murashige & Skoog, 1962) and MS containing various concentrations of NAA and IBA (Fig. 4A and Table 3). Rooting was performed on 56 explants (seven replicates with eight explants). After 4 weeks of rooting, the microshoot percentage, root length and number of roots were counted. Plantlets with well-developed roots (Fig. 4B) were placed in Jiffy-7-Pellet® (Fig. 4C). Subsequently, once the roots escaped from the Jiffy, the plants were placed in plastic pots containing a mixture of peat and perlite (75% and 25%, respectively) (Fig. 4D). To favour hardening, the plantlets were kept at 25 ± 1 °C with 16 h of light and 8 h of darkness.

Figure 4 Hieracium lucidum subsp. lucidum.

(A) Shoots in hormone-free MS basal medium including vitamins; (B) In vitro rooting of shoots; (C) plantlet placed in Jiffy-7®; (D) plantlet in plastic containers containing a mixture of peat and perlite.

Table 3 Plant growth regulators used in the growth media to induce the regeneration of roots from Hieracium lucidum subsp. lucidum.

Medium	PGR	
IBA mg L−1¹	NAA mg L−1	
MS0	0	0	
R1	0.5	0	
R2	1	0	
R3	0	0.5	
R4	0	1	
Note:

MS0, control, hormone-free; R1, R2, medium with different concentration of Indole-3-butyric acid (IBA) at 0.50 and 1 mg L−1 respectively; R3, R4, medium with concentration of Naphthaleneacetic Acid (NAA) 0.50 and 1 mg L−1 respectively.

Statistical analysis of in vitro propagation tests

In this experiment, a completely randomised design was conducted to test the effect of medium on callus, shoot regeneration and rooting. A two-way analysis of variance (ANOVA) was conducted to identify statistically significant disparities and potential interactions among the PGRs (with a significance level of P ≤ 0.05) (Pereira et al., 2018). When the interaction between two factors lacked significance, a one-way ANOVA was executed, individually analyzing each factor, and determining mean distinctions through Tukey’s test. The statistical analysis was carried out utilizing SYSTAT 13 software.

Results

Seed germination tests

Germination began in all tests within 1 week. The summary analysis of variance for the adopted model (see File S1) shows that only the effect of the different temperatures on seed germination can be considered as significant (P-value = 0.005), being 15 °C the best experimented value. Conversely, the effect of the tested solutions cannot be considered significant (P-value = 0.645). The highest germination percentage (95%) and germination speed (T50: 8.2, MTG: 8.6) occurred at a constant temperature of 15 °C in full darkness. Then there is a decrease in germination as the temperature increases (−5% in pure water at 20 °C and −10% in pure water at 25 °C). Alternating the temperature (30/15 °C) dropped the germination percentage to 70%. The results obtained from germination tests conducted on Hieracium lucidum subsp. lucidum seeds collected in 2020 and 2021 showed an absence of dormancy and a high germination capacity of freshly harvested seeds (Tables 4 and 5). Germination tests carried out in 10−3 M gibberellic acid (GA3) provided similar or slightly higher germination percentages, with a maximum difference of 5% (corresponding to one seed). The seeds of H. lucidum do not have particular systems that allow them to penetrate into the soil, but remain deposited on its surface. These results demonstrate that these seeds take advantage of the protection offered by pockets of soil in rock cracks during germination. Seeds that germinate in the open field have less chance of germinating. The temperature of 15° occurs in late autumn ensuring more constant water availability due to rainfall.

Table 4 Summary table of the values of the germination indices obtained from seeds collected in 2020 and 2021.

Germination	Temperature	Photoperiod
(light/dark)	T1 (d)	T50 (d)	Tmax (d)	MTG (d)	
2020	
95%(1)	Constant 15 °C	0/24 h	5	8.2	15	8.6	
95%(2)	Constant 15 °C	0/24 h	5	8.1	15	8.4	
90%(1)	Constant 20 °C	0/24 h	6	10	16	10.8	
95%(2)	Constant 20 °C	0/24 h	5	8.1	16	10.4	
80%(1)	Constant 25 °C	0/24 h	7	11.2	21	12.2	
80%(2)	Constant 25 °C	0/24 h	6	10.4	20	11.4	
70%(1)	Alternating 30/15 °C	16/8 h	7	12.5	22	13.8	
75%(2)	Alternating 30/15 °C	16/8 h	6	10.7	20	11.7	
2021	
90%(1)	Constant 15 °C	0/24 h	5	9.2	15	10.2	
90%(2)	Constant 15 °C	0/24 h	5	9.2	15	9.6	
85%(1)	Constant 20 °C	0/24 h	6	10.6	18	11.7	
85%(2)	Constant 20 °C	0/24 h	5	10	16	10.7	
80%(1)	Constant 25 °C	0/24 h	6	11.2	20	12.3	
80%(2)	Constant 25 °C	0/24 h	6	10.2	19	11.1	
70%(1)	Alternating 30/15 °C	16/8 h	7	12.3	21	13.1	
75%(2)	Alternating 30/15 °C	16/8 h	7	11.5	19	12.6	
Notes:

Percentage of germination obtained in pure water(1) and in 10−3 M gibberellic acid (GA3) water solution(2).

T1, germination delay in days; T50, median germination time in days; Tmax, maximum germination time in days; MTG, mean time to germination in days.

Table 5 Statistics of the values of the germination indices obtained from seeds collected in 2020 and 2021.

Variable	Germination % mean	
Media	
PW	82.50	
GA3	84.40 non-significant	
Temperatures		
T15	92.50 a	
T20	88.75 a	
T25	80.00 b	
TA	72.50 b	
Notes:

PW, pure water; GA3, 10−3 M gibberellic acid water solution; T15, temperature constant at 15 °C; T20, temp. constant at 20 °C; T25, temp. constant at 25 °C; TA alternating temp. 30/15 °C.

The different letters grouped indicate statistically significant differences between each temperature tested (Tukey’s test, P ≤ 0.05).

Vegetative propagation test

The vegetative propagation test (Table 6) showed that 2 months after planting, the cuttings that received the IBA treatment and were placed in the rooting medium (50% sand-perlite and 50% peat) were rooted at 100% and produced new leaves after rooting. The control cuttings grew in the rooting medium with 50% sand-perlite and 50% peat rooted approximately 3 months after planting, but only 50% produced new leaves after rooting. Furthermore, they formed fewer roots (3 vs 6) with shorter lengths (3 vs 5 cm) than those produced by the IBA-treated cuttings. Additionally, the number of leaves produced after rooting was lower in the control cuttings (7 vs 12). All cuttings growing in rooting medium consisting of 50% fine sand and 50% sawdust rapidly dried.

Table 6 Effect of hormonal treatment and rooting medium on vegetative propagation of Hieracium lucidum subsp. lucidum.

Treatment	No. samples	No. of success	Rooting time (months)	No. of roots (mean)	Mean length of roots (cm)	No. of leaves	
IBA + RM1	6	6	2	6	5	12	
IBA + RM2	6	0	–	–	–	–	
Control + RM1	6	3	3	3	3	7	
Control + RM2	6	0	–	–	–	–	
Note:

IBA, talc formulation of Indole-3-butyric acid (IBA); RM1, rooting medium 1, 50% sand-perlite and 50% peat; RM2, rooting medium 2, 50% of fine sand and 50% sawdust.

In vitro propagation test

The leaf explants responded differently to different concentrations of BAP and mT in combination with NAA and 2,4-D. Increased callus production and shoot regeneration were observed when mT was present in the culture medium at concentrations of 1 and 2 mg L−1 in combination with 2,4-D and NAA (1 mg L−1). Maximum callus production occurred when 2 mg L−1 of mT was present in the medium in combination with 1 mg L−1 of 2,4-D (86% of callus). Leaf explant culture on medium without plant growth regulators (MSO) showed no growth. The callus rate was significantly reduced in the culture medium enriched with BAP (1 or 2 mg L−1) in combination with NAA and 2,4-D.

When mT was replaced with 2 mg L−1 BAP in combination with 1 mg L−1 of 2,4-D, a lower callus rate of 39% was observed, and this rate decreased more until a percentage of callus equal to 13% was reached in the medium when BAP was combined with NAA (Table 7).

Table 7 Effect of different auxins on in vitro regeneration of callus after four weeks of culture and regeneration of shoots after seven weeks of culture of Hieracium lucidum subsp. lucidum.

Medium	Percentage of explants producing callus	Numbers of shoots per explants	
MS0	0	0	
B1	13	3.86 ± 0.4 f	
B2	21	4.14 ± 0.3 ef	
B3	30	4.57 ± 0.2 e	
B4	39	4.86 ± 0.4 e	
mT1	49	6.29 ± 0.3 d	
mT2	51	7.57 ± 0.7 c	
mT3	74	11.29 ± 0.5 b	
mT4	86	14.43 ± 0.5 a	
Note:

Value represent mean ± SE. Medium acronyms refer to Table 2. The different letters grouped for each single column indicate statistically significant differences between each culture medium tested (Tukey’s test, P ≤ 0.05).

The addition of BAP (1 or 2 mg L−1) in combination with NAA and 2,4-D significantly reduced the rate of shoot production (Fig. 4). When mT was replaced with 2 mg L−1 BAP in combination with 1 mg L−1 of 2,4-D, a mean number of buds of 4.86 per explant was observed, and this number decreased to an average of 3.86 shoots in the medium with BAP and NAA (Table 8).

Table 8 Effect of different auxins on in vitro rooting of Hieracium lucidum subsp. lucidum, after four weeks of culture.

Medium	Percentage of explants producing root	Numbers of roots per explants	Root length	
MS0	79	9.41 ± 0.5 a	1.49 ± 0.0 a	
R1	60	6.11 ± 0.5 c	0.73 ± 0.1 c	
R2	67	8.59 ± 0.3 b	0.93 ± 0.0 b	
R3	34	4.43 ± 0.4 d	0.54 ± 0.0 d	
R4	41	3.86 ± 0.3 d	0.74 ± 0.0 c	
Note:

Value represent mean ± SE. Medium acronyms refer to Table 3. The different letters grouped for each single column indicate statistically significant differences between each culture medium tested (Tukey’s test, P ≤ 0.05).

The regenerated shoots from the leaf explants grown on shoot regeneration media were transferred to root regeneration media. After four weeks of culture, root induction gradually decreased when IBA was replaced with NAA at the same concentration in the media (Table 7). In MS basal medium with vitamins (Murashige & Skoog, 1962) and without PGRs, better root induction was observed both in terms of percentage (79%), number of roots per explant (9.4) and root length (1.49 cm). The roots were long, abundant and vigorous in appearance. The MS basal medium enriched with IBA 1 mg L−1 showed a better response (67%), with a maximum number of roots per shoot of 8.59 and a root length of 0.93 cm compared to the medium in which IBA was substituted with NAA at the same concentration (Fig. 4).

The rooted plantlets with fully expanded leaves were placed in Jiffy-7-Pellet®, and once the roots had escaped from the Jiffy, the plantlets were placed inside plastic containers containing a mixture of peat (75%) and perlite (25%). Once hardened, the plantlets were planted in pots containing garden soil and maintained in a greenhouse. A survival rate of 90% was obtained in greenhouse conditions where good growth and a high degree of homogeneity without visible variations were observed. A total of 50 acclimated plants were produced.

Discussion

The aims of this study were to develop efficient protocols for seed germination and vegetative and in vitro propagation to improve applied knowledge about the ex situ conservation of critically endangered Sicilian endemic Hieracium lucidum subsp. lucidum. For this purpose, germination assays, vegetative propagation and in vitro tests were accomplished as part of an integrated investigation.

The results obtained from the germination tests showed that seeds of H. lucidum subsp. lucidum had no dormancy and a high germination capacity (70–95%) immediately following harvest. This adaptation allows populations living in vertical cliffs to have an immediate availability of new individuals under unfavourable ecological conditions and thus greater opportunities for affirmation of the progeny (Di Gristina, Varshanidze & Domina, 2020).

Among other things, as evidenced by monitoring the population of H. lucidum subsp. lucidum in nature, its seedlings have low competitive capacity, and those emerging at the base of the cliffs are quickly overwhelmed by the surrounding vegetation and destined to perish. This last phenomenon plays a fundamental role in limiting the expansion of the species outside its primary habitat.

The highest percentage of germination (95%) was obtained at a constant temperature of 15 °C in the dark. However, high germination rates were also obtained at all tested thermoperiods, although they decreased as the temperature increased, reaching a minimum of 70% at an alternating temperature of 30/15 °C. Based on these results, H. lucidum subsp. lucidum germinates better at cool or medium constant temperatures. The germination tests conducted in GA3 did not show substantial variations from those conducted in pure water; therefore, the seed germination of H. lucidum subsp. lucidum was not influenced by GA3. No other data on germination are available for this taxon.

Regarding the vegetative propagation test, the total percentage of rooting cuttings obtained in the shortest time, the greater number of roots and their greater length, and the greater number of leaves emitted after rooting suggest that treatment with IBA is beneficial for rooting H. lucidum subsp. lucidum cuttings. However, a larger-scale trial is required to confirm this observation.

Vegetative propagation tests also showed that the best rooting medium consisted of 50% sand-perlite and 50% peat. Overall, the results obtained from the rooting cuttings can be defined as encouraging, considering the low productivity capacity of mature H. lucidum subsp. lucidum seeds found in nature (about 10–15 per capitulum). Vegetative reproduction could be a useful multiplication technique for the ornamental use of this taxon.

A highly efficient in vitro micropropagation protocol was developed using leaf explants. The effect of the combination of two different cytokinins, such as BAP and mT, combined with NAA and 2,4-D was examined for the regeneration of callus and shoots. Callus formation was observed in leaf explants incubated on MS basal medium with vitamins (Murashige & Skoog, 1962) and different auxin/cytokinin combinations after 4 weeks of incubation and shoot regeneration after 7 weeks. The induction rates of callus and shoots varied depending on the combination of the applied PGRsgrowth regulators; mT is an alternative to benzyladenine (BA), zeatin (ZEA), and kinetin (KIN) in plant tissue culture (Aremu et al., 2012). It has been used to increase the in vitro plant propagation efficiency in Citrus (Chiancone et al., 2015), Capparis (Gianguzzi et al., 2019) and Carthamus (Vijayakumar et al., 2017). The direction of organogenesis is regulated by the type, ratio and concentration of cytokinins and auxins (Xi et al., 2013). Cellular recognition and/or mechanism of action of cytokinins may affect the different responses to PGR combinations (Kim et al., 2001). In the present study, when mT was present in the culture medium at a concentration of 1 or 2 mg L−1 in combination with 1 mg L−1 2,4-D and NAA, there was a greater production of callus compared to media in which mT was replaced with BAP. Maximum callus production was observed when 2 mg L−1 mT was present in the medium in combination with 1 mg L−1 2,4-D (86%) with an average number of shoot regenerations equal to 14.43. Despite hormone-free medium stimulated callus formation on the shoots and plantlets of explants in Lilium (Kedra & Bach, 2005). In our experiments callus induction and shoot regeneration did not occur in the hormone-free medium but occurred at all combinations at different percentages and numbers of shoots. Leaf cultures of other Asteraceae, such as Adenostyles alpina subsp. macrocephala (Huter, Porta & Rigo) Dillenb. & Kadereit (Gianguzzi et al., 2023), Carlina acaulis L. (Trejgell, Dąbrowska & Tretyn, 2009), Echinacea purpurea (L.) Moench (Korach et al., 2002) and Eclipta alba (L.) Hassk. (Dhaka & Kothari, 2005), benefit from the combined effect of cytokinins and auxins, and the highest multiplication rate was achieved with supplementation with BA MS medium. The response of the auxin–cytokinin ratio in organogenic differentiation has also been documented for Bignoniaceae, Hypericaceae and Piperaceae (Lisowska & Wysokinska, 2000; Pereira et al., 2000; Pretto & Santarém, 2000).

From the obtained results, a decrease in the percentage of callus and the number of shoots per explant was observed after substitution in the medium with BAP instead of mT at the same concentration. Of the two concentrations of BAP tested in combination with NAA and 2,4-D, the 2 mg L−1 BAP in combination with 1 mg L−1 of 2,4-D induced a higher percentage of callus (39%) with an average number of shoots per explant of 4.86. A response percentage lower than 13% was observed in the medium containing 1 mg L−1 BAP in combination with 1 mg L−1 of NAA. The callus-regenerated shoots were harvested and transferred to rooting media. The response in terms of rooting percentage, number and length was evaluated after 4 weeks. The best results were recorded in the hormone-free MS basal medium and in the medium with IBA. Root regeneration is a crucial step during plant micropropagation. IBA is preferably used for in vitro root regeneration (Deklerke, Terbruyge & Marinova, 1997; Ramesh et al., 2005). In the present study, the maximum rooting percentage (79%) was obtained in MS basal medium without hormones, with 9.4 roots per explant and an average length of 1.49 cm. Among the solutions tested (IBA at 0.5, 1 mg L−1 or NAA at 0.5, 1 mg L−1) the highest root production occurred with the presence of IBA at a concentration of 1 mg L−1. IBA is one of the most commonly used auxins to induce root formation by organogenesis. The efficacy of IBA for rooting in regenerating shoots has been reported for several species, such as Adenostyles alpina subsp. macrocephala (Gianguzzi et al., 2023) and Capparis spinosa L. (Gianguzzi, Barone & Sottile, 2020). The acclimation step was successfully performed with Jiffy-7® pellets. Subsequently, the plants were placed in pots with a mixture of peat and perlite (75% and 25%, respectively). Once hardened, the plantlets were planted in pots containing garden soil, maintained in the greenhouse for 2 months, and finally in open air, ready for reinforcement activities of the natural population.

Conclusions

The carbonate cliffs are an important refuge for plant biodiversity in the Mediterranean, as they are the only truly natural environments left in this geographical area that have been so strongly anthropised since the dawn of agriculture. All possible actions should be taken to preserve the environment and the plant species living there.

This study is the first integrated report of seed germination and vegetative and in vitro propagation for critically endangered Sicilian endemic Hieracium lucidum subsp. lucidum.

We examined the different propagation techniques of this taxon as a case study that can also be adapted to the conservation of other species that share the same conservation problems.

Seed germination does not appear to be controlled by temperature; in fact, a high germination capacity was obtained at all tested temperatures. The seeds had no dormancy, and the germination process was not influenced by giberellins.

A satisfactory vegetative reproductive capacity appeared from the number of rooted stem cuttings obtained. The application of a talc formulation of IBA to the basal portion of the cuttings improved rooting compared to untreated cuttings.

This research confirmed that leaves have great organogenic potential for shoot formation for in vitro propagation; however, the response is directly related to the combinations of growth regulators. Maximum regeneration of the callus and shoots was achieved in the medium containing 2 mg L−1 of mT and 1 mg L−1 of 2,4-D, and maximum regeneration of roots was induced in hormone-free MS medium or containing 1 mg L−1 of IBA. In vitro micropropagation provides an efficient method for the regeneration of adventitious shoots from H. lucidum leaf explants that will be useful for species conservation.

Based on the obtained results, it can be stated that H. lucidum subsp. lucidum can be successfully propagated using one of the tested techniques reported in this study, depending on the availability of the material for reproduction, thus contributing to the conservation of this critically endangered element of the Sicilian flora. Propagation by seed allows the genetic variability of the populations, but the studied species is not able to produce fertile seeds in quantities suitable for multiplication programmes. Vegetative and in vitro reproduction can be used to support propagation by seed to have a large number of plants in a short time but do not determine genetic variability. In vitro reproduction resulted in the most effective multiplication technique, based on the number of plants produced in a relatively short time. It is essential that propagation material be collected from several individuals to preserve the highest genetic variability.

Supplemental Information

Supplemental Information 1 Binary Logistic Regression: Germination versus Temperature; Solution.

Click here for additional data file.

Supplemental Information 2 Raw data for Table 4.

Click here for additional data file.

Supplemental Information 3 Raw data for Tables 7 & 8.

Click here for additional data file.

Our sincere thanks to our colleague prof. Stefano Barone (University of Palermo) for the discussions and suggestions regarding the design and analysis of the seed germination experiments.

Additional Information and Declarations

Competing Interests

Author Contributions

Data Availability

The authors declare that they have no competing interests.

Valeria Gianguzzi conceived and designed the experiments, performed the experiments, analyzed the data, prepared figures and/or tables, authored or reviewed drafts of the article, and approved the final draft.

Emilio Di Gristina conceived and designed the experiments, performed the experiments, analyzed the data, prepared figures and/or tables, authored or reviewed drafts of the article, and approved the final draft.

Giulio Barone conceived and designed the experiments, performed the experiments, analyzed the data, prepared figures and/or tables, authored or reviewed drafts of the article, and approved the final draft.

Francesco Sottile conceived and designed the experiments, analyzed the data, authored or reviewed drafts of the article, and approved the final draft.

Gianniantonio Domina conceived and designed the experiments, analyzed the data, authored or reviewed drafts of the article, and approved the final draft.

The following information was supplied regarding data availability:

The raw data are available in the Supplemental Files.

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
