# Peer review of "Seed germination and vegetative and in vitro propagation of Hieracium lucidum subsp. lucidum (Asteraceae), a critically endangered endemic taxon of the Sicilian flora"

_PeerJ, doi:10.7717/peerj.16839_

## Round 0.1 · original submission · Major Revisions

Endangered plants have special importance in agricultural and non-agricultural fields due to the sustainability of agro-ecosystems. Each study that is conducted to understand the germination and propagation of this plant is important to survive this plant. Therefore, this article can provide some important knowledge to conserve Hieracium lucidum subsp. lucidum (Asteraceae) populations. However, this article needs a major revision.

Your article needs linguistic revision. I suggest you should get help from one of your colleagues or our editing service to edit your article.

Please follow the suggestions of all reviewers. If you cannot accept one or more of their suggestions, you should explain the reason why you refused it.

Please check your references and the reference list.

The methodology should be detailed to make it reproducible.

**Language Note:** The Academic Editor has identified that the English language must be improved. PeerJ can provide language editing services - please contact us at copyediting@peerj.com for pricing (be sure to provide your manuscript number and title). Alternatively, you should make your own arrangements to improve the language quality and provide details in your response letter. – PeerJ Staff

Reviewer 1 ·

Basic reporting

In this manuscript Gianguzzi et al. reported their experience in the propagation (via seeds and in vitro) of the Critically endangered Hieracium lucidum subsp. lucidum a species endemic to Sicily. They tested both seed germination, rooting capacity, and in vitro propagation from leaves portions. The topic is interesting and strategies for propagation and conservation of highly threatened species are of relevant interest. However, I believe that the manuscript needs to be improved in readability. Moreover, I think that the authors do not discuss in depth the implications for Hieracium lucidum conservation of their results.
Finally, I believe that the germination data are only described and not statistically analysed not capitalising enough the full potential of obtained results. I suggest to analyse the germination data using GLM and GLMM to fully understand which is the most effective germination protocol for seed germination.
Below some other comments to the specific sections.

Experimental design

Mat & Methods
Line 164-165: why do you use these temperatures? There is a connection with the temperature recorded in the field from weather stations or data loggers or do you choose for other reasons? The same apply to alternate temperatures, why 30/15°C? Please explain.
Moreover, it is not clear where do you test germination requirements. In growth chambers? In incubators? Which was the light intensity of lamps?

Validity of the findings

Introduction
In my opinion the introduction could be improved, and it is necessary to make it more linear and fluid. For example, until line 100 the authors report information on studies carried out so far in Sicilian Hieracium. Then they discuss the importance of in vitro propagation and subsequently they return to talk about Sicily and endemic Hieracium of the island, including the species that is the focus of this study. I suggest moving the section on in vitro propagation at the end of the introduction.

Results
Seed germination tests
It is not clear how you test seed germination at different temperatures and treatments. It seems to me that only descriptive summary is reported, however I expected to find some statistical analysis (e.g.: GLM o GLMM) to test differences among temperatures, years and treatments (with and without GA3). Moreover, in line 255-256 the authors stated “By increasing the temperature up to 30°C, the percentage of germination dropped (70%)”; however, in Mat & methods they say to have use only 15-20-25°C. Where does this 30°C appear from?

Additional comments

Discussion
The discussion is long and repetitive. In my opinion could be shortened, but I suggest to add a section on implications for the species conservation. Do you believe that produced plants could be easily used to reinforce wild populations? Or do you think that more hardening passages are required before reintroduction? How many individuals could be produced with this technique? Which is according to the author the advantages and disadvantages of in vitro propagation compared to seed germination? I think that a short discussion on these points could be of interest for readers.

Conclusions
This section is too long. In my opinion the authors should directly focus on the results of the paper and implication for conservation of Hieracium lucidum and not spent words on habitat of the species, importance of micropropagation and aim of the study.

Minor comments

Line 130: “Mt Gallo although is a regional natural reserve almost yearly is affected by wildfires that which also reach the vertical walls” I suggest rephrasing this sentence.
Line 163: change and with an
Please check the figure order, Fig. 4 appear earlier in the text than Fig. 3. Moreover, I believe that Fig. 3 could be removed (less useful to understand data) or better replaced by a graph for example that show germination percentage bar with statistical analysis.

·

Basic reporting

In some sections, the number of references is so high, and this makes the paper hard to follow for readers. I believe you have conducted thorough research, but the low level of English usage makes it a bit difficult to follow.

Experimental design

The experimental design that was used by the authors is the right choice for a germination test in Petri dishes medium.

Validity of the findings

no comment

Additional comments

Seed germination, vegetative and in vitro propagation of Hieracium lucidum subsp. lucidum (Asteraceae), a critically endangered endemic taxon of the Sicilian flora (#89344)

Comment 1: It would be better to shorten the abstract to a total of 250 words.

Comment 2: In line 59, what is your point in mentioning that "The germination process is not influenced by Gibberellins"?

Comment 3: In formal writing, it is generally better to write "two months" as words rather than using the numeral "2" for the number.

Comment 4: In line 70, it is better to say, "medium enriched with..." The English in the abstract does not meet the required standards of clarity and precision.

Comment 5: In line 89, you mentioned several endemic taxa with very local distribution. This is interesting information. As a suggestion, you can present this information in the form of a table to reduce the number of sentences containing references, making the paper easier to follow.

Comment 6: In line 161, it's better not to start the sentence with a number. Instead, you would write: "A total of 80 seeds…"

Comment 7: In line 246, what was your reason for using Tukey's test in means comparison?

Comment 8: The English level of the entire manuscript does not meet the required standard. In some sentences, you have simply put together different sentences without using any common connectors in English. The English quality of the manuscript should be improved for publication.

Comment 9: In the reference section, to help readers access specific references or allow Journal Editors to verify the references, please provide the DOI as a direct link for all references if available. For example:
Mráz P, ZdvoYák P, 2019. Reproductive pathways in Hieracium s.s. (Asteraceae): strict sexuality in diploids and apomixis in polyploids. Annals of Botany 123:391–403. DOI: 10.1093/aob/mcy137

This is not an appropriate reference for accessing this paper. Instead, you can use the DOI as a web link, like this:
https://doi.org/10.1093/aob/mcy137
and place it at the end of your paper:

Mráz P, ZdvoYák P, 2019. Reproductive pathways in Hieracium s.s. (Asteraceae): strict sexuality in diploids and apomixis in polyploids. Annals of Botany 123:391–403. https://doi.org/10.1093/aob/mcy137

You can also use Mendeley software to manage your references.

Comment 10: Please provide a full explanation of each medium that comes in abbreviation form for all the tables. Explain what MS0, MT1, and others are.

Comment 11: In table 3, what was your purpose in using a comma (,) between values? For example, you use 8,2. I think it should be 8.2?

Reviewer 3 ·

Basic reporting

Carefully reading the paper before submission, could have avoided many remarks, ranging from professional remarks (e.g. Petri is always written with uppercase) to misused common words (e.g. interested instead of benefited or similar – line 126: or punctiform instead of punctual, in line 127) and, specially, syntax problems (e.g., lines 129 and 130). These flaws are signalized on the text, most of the time with suggestions.

Not being a review paper, I find that it has too many references (around 70) and some are only referenced once and along with other relevant works. I believed that, at least those can be screened and reevaluated regarding its relevancy. That´s the case of Bengi & Yeald, 2005, and Trejgell, Michalska & Tretyn, 2010 in the following sentence: “Many threatened species can be quickly propagated and preserved with this technology by using a minimum of plant material and reducing the impact on populations (Bengi & Yeald, 2005; Trejgell, Michalska & Tretyn, 2010; Gianguzzi et al., 2023).” but many others can be found.

Experimental design

I found that experimental design of the In vitro propagation test procedure is not clearly explained. I therefore suggest to the authors:
1. To provide a scheme with the experimental design of the entire In vitro propagation test procedure (including the n of the samples);
2. To separate what was done for callus and for shoots, in both Materials & Methods and Results sections – making it more readable for those who only want to know about roots or about shoots.
3. To use this scheme, along with the comments/suggestions made by the reviewers to rewrite this part, aiming at making it clearer and reproducible.

Validity of the findings

This study is very important because it is “the first integrated report of seed germination, vegetative and in vitro propagation for the critically endangered Sicilian endemic Hieracium lucidum subsp. lucidum”. The improvement of the in vitro propagation tests experimental design description is, however, fundamental to evaluate the validity of the findings. Regarding the other tests, seed germination and the vegetative propagation tests conclusions (e.g. lines 432-434) are sometimes too affirmatives when they should be more cautious. More detailed revision can be found in the paper itself. Samples size should be included always everywhere, because this is a measure of data robustness.

Annotated reviews are not available for download in order to protect the identity of reviewers who chose to remain anonymous.

---

## Round 0.2 · Minor Revisions

I appreciate your positive approach in considering the reviewers' recommendations. To enhance your article further, I recommend carefully reviewing and incorporating the suggestions provided by the reviewers. Please take the time to thoughtfully organize your article based on these valuable insights. Thank you for your dedication to refining your work.

Reviewer 1 ·

Basic reporting

I wish to appreciate the effort provided by the authors to address my comments and present a revised version of the paper. I still have some comments regarding the germination tests (see below) that the actual version do not contribute to clarify.
Moreover, the statistical analysis applied to germination tests are not appropriate. I suggest to the authors this manuscript of Sileshi (2012) "A critique of current trends in the statistical analysis of seed germination and viability data" Seed Science and Research 22: 145-159 https://doi.org/10.1017/S0960258512000025
Sileshi highlight that "ANOVA results can be invalid if the number of replicates is small". Among the more powerful and flexible alternatives to ANOVA and NPARTs are LMMs, GLMs and GLMMs (as I suggested in my first review round).

Moreover in the results there are no summary of the analysis, but only a general description (Table 4 and 5).

In your tests you used may variables (Temperatures, GA3, Year) and the effects of all these variables cannot be disentagled only with a simple ANOVA.

Experimental design

In the first version you reported the number of seeds tested and the numebr of replicates (20 x 4 replicates). In this version this information is omitted, although I believe is extremely important. Moreover, I supposed you tested 20 seeds in 4 replicated for any of the temperature reported both with and without GA3 (then you used 640 seeds per year); did I get it right? Please clarify.
Moreover, if I well understood you tested constant temperature in continuous darkness and alternate temperatures with light. Why you choose this protocol?
Finally, you apply the same protocol both in 2020 and 2021?

Validity of the findings

No comment

·

Basic reporting

The authors of the paper have completely revised all of my comments and the final version of the manuscript does not have any major issues. Have a great week.

Experimental design

The authors have revised this section.

Validity of the findings

I confirm the validity of findings.

Reviewer 3 ·

Basic reporting

The readability of the article has improved greatly in this second version, and so has its clarity. The abstract benefited from the reduction, as did the references.

Experimental design

The research question is well defined.

Validity of the findings

This study provides a suitable technical itinerary for replication in similar cases.

Annotated reviews are not available for download in order to protect the identity of reviewers who chose to remain anonymous.

---

## Round 0.3 · accepted · Accept

Thank you for accepting the suggestions of the reviewers and improving your article. I have a few minor edits that you need to address while in production:

Please check the Latin names of plant species and their authors
Please remove one dot from two dots (..)
Please write the temperatures with their accuracy (like 25 ± 1°C)
Please remove the acronym “G” and its explanation from the legend of Table 4.
Please check style considerations (like g/L- ¹)

The article is now acceptable.

Reviewer 1 ·

Basic reporting

The authors have revised the manuscript according to my major comments regarding statistical analysis of seed germination, and as expected results changed compared to the first version.
In my opinion the manuscript have no other major issues and is now suitable for publication.

Experimental design

no comment

Validity of the findings

no comment